# Production Planning to Reduce Production Cost and Formaldehyde Emission in Furniture Production Process Using Medium-Density Fiberboard

**Taeho Kim**

Department of Business Administration, Incheon National University, 119 Academy-ro, Yeonsoo-gu, Incheon 22012, Korea; thkim@inu.ac.kr; Tel.: +82-10-2614-4940

**Abstract:** This research seeks to improve the production process in the Korean furniture industry by reducing the amount of medium-density fibreboard, that is commonly used to produce furniture, in order to reduce production costs and formaldehyde emissions. This research selects a representative company from the Korean furniture industry to examine its optimal amount of medium-density fibreboard used, using the variables of a previous company; the sale levels, the Korea National Productivity Award Index, and technical efficiencies obtained from a previous study. By using its 2016 production level, we compare it with the amount of medium-density fibreboards actually used in 2016, and apply the results to the entire Korean furniture industry. In conclusion, the Korean furniture industry can minimize the amount of medium-density fibreboards used without reducing current production levels, and thereby save production costs, and contribute to substantially reducing formaldehyde emissions.

**Keywords:** furniture production process; medium density fiberboard; production cost; formaldehyde emission; two-dimensional bin packing/cutting stock problem

---

## 1. Introduction

The furniture production industry has been a key industry in Korea since 1970, when Korea started to develop its economy after the Korean War. The total sales of the Korean furniture industry were US $982,142,857, with 11,572 companies and 66,932 workers engaged, which shows an increase of 2000 companies and 4000 workers, since 2003 [1]. The main reason for this expansion is demand growth, due to Korea's rapid urbanization, and the opening of the furniture market to global companies.

However, as most companies have focused on the domestic market, rather than global markets and global brands, companies such as Pottery Barn, IKEA, and Ethan Allen have entered the Korean market, the Korean furniture industry has since been exposed to excessive competition, and many companies have risked failure [2]. Moreover, although the needs and wants of Korean customers have diversified, Korean companies cannot produce furniture that reflects consumers' various wants and needs, because they lack sufficient technology and capital, due to small firm sizes [3,4]. Recent interest in environmental hormones emitted from furniture has led firms to use expensive natural wood panels rather than artificial, wood-based panels, such as medium-density fibreboard (MDF) and particle board (PB), which are relatively cheap and easy to process.

Furniture is produced with wood-based panels and other materials, such as plastics, steel, and fabric. More than 95% of furniture, however, is made of wood-based panels. Wood-based panels for furniture production are classified into natural wood, in furniture production [5]. While natural wood was originally used to produce furniture, it has some weaknesses in that it is expensive and cannot be easily processed in various shapes. Above all, world forests may have been depleted from

the use of natural wood for furniture production. Thus, artificial wood-based panels, such as PB and MDF were invented and have been used broadly for furniture production. PB was invented in 1887 in Germany, under the name, "artificial wood", and it has long been used for office furniture and kitchen furniture (https://en.wikipedia.org/wiki/). MDF was invented in 1965 in the U.S.A. and was industrialized for the production of all kinds of furniture in the 1970s [5]. Both types of artificial wood are more easily processed into various shapes, more resilient against shocks than natural wood, and contribute to environmental protection measures because they are produced with discarded or recycled woods. Due to these advantages, the global production and consumption of PB and MDF have grown steadily since 1980. The consumption of MDF has increased by an average of 10% annually since 1980. It reached 48,000,000 $m^3$ in 2006 and 96,400,000 $m^3$ (US $3,850,000,000 in value) in 2016 [4]. Of this, approximately 60% is used to produce furniture, and 40% is used for construction. China, Turkey, Brazil, and Poland are the world's key MDF-producing countries and account for 59%, 5%, 4%, and 4% of MDF output, respectively in 2016 [4]. The statistics for PB are similar. Global consumption in 2016 was 118,966,000 $m^3$, an increase of 41% from 2001, and most PB is used for furniture production [5]. China, the U.S.A., Canada, Russia, and Germany are the world's largest PB producing countries and account for 19.4%, 13.8%, 8.1%, 6.2%, and 5.9% of PB output, respectively, in 2016 [4].

Korea is also an important country in both the production and consumption of MDF and PB because the nation's furniture and construction industries have grown steadily since the 1970s. Three factories, owned by 3 companies, produced 816,000 $m^3$ of PB, and 10 factories, owned by 6 companies, produced 1,859,000 $m^3$ of MDF in 2016. Korea's MDF production capacity is ranked ninth in the world. A total of 1,185,000 $m^3$ of MDF and 2,070,090 $m^3$ of PB were used for furniture production in Korea in 2016, in addition to the 1,275,000 $m^3$ of PB and 116,000 $m^3$ of MDF that was imported that year [5]. Due to these productions, MDF and PB are now the main factors affecting the production cost and product liability of Korean furniture manufacturers. Table 1 illustrates the production amount of PB and MDF of important production countries in 2016.

**Table 1.** Summary of world production information of particle board (PB) and medium-density fibreboard (MDF) in 2016.

| MDF ($m^3$) | | PB ($m^3$) | |
|---|---|---|---|
| China | 56,876,000 | China | 23,079,404 |
| Turkey | 4,820,000 | The U.S.A. | 16,417,308 |
| Brazil | 3,856,000 | Canada | 9,636,246 |
| Poland | 3,856,000 | Russia | 7,375,892 |
| Korea | 1,859,000 | Germany | 7,018,994 |
| | | Korea | 816,000 |

While PB and MDF are good for producing furniture, they also have a critical weakness in their emission of formaldehyde, which is a kind of toxic chemical and environmental hormone. Both PB and MDF emit formaldehyde mainly because of adhesives used to amalgamate ground wood, as chemicals are rarely used during the grinding and hardening processes in wood. In other words, the main source of formaldehyde, emitted from PB and MDF, are adhesives, which are used to amalgamated ground wood to make it into the shape of a board. Formaldehyde can stimulate human eyes and skin conditions and cause respiratory diseases, atopy, and cancers after long term exposure. Many countries have designated formaldehyde as a dangerous toxic chemical and regulate its use (http://blog.daum.net/furniture6025/8747739). MDF of other classes are still used broadly. It is classified it into four groups based on its emissions levels, although the use of top-level MDF or the reduction of MDF use [6], and MDF producers are encouraged to use adhesives and resins with lower emission levels of formaldehyde, thereby fulfilling the requirement of super E0 class [6].

This research focuses on helping Korean small and mid-size furniture manufacturers to use MDF to find a way to rehabilitate and contribute to environmental protection, by analysing their

production processes, and suggesting ways in which they can save on production costs and protect the environment. The amount of furniture produced using MDF cannot be controlled by furniture companies. Customers or environmentalists can do that. Manufacturers must contribute to reducing formaldehyde in the air by reducing the amount of MDF used in production processes, as well as in limiting or eliminating furniture using this chemical. This means that a reduction in the total amount of MDF existing in Korea (either as raw material or as furniture) results in less total amount of formaldehyde in the air. The reason we selected MDF over PB as the target wood-based panel for this research, is that Korea produces much more MDF than PB domestically, which implies that Korean furniture manufacturers can adjust their procurement of MDF more easily than that of PB. As noted above, most MDF is produced domestically, while most PB is imported into Korea.

For this study, we first selected a representative company from the Korean furniture industry, based on their sales, management quality, and technical efficiencies. Sales demonstrate the scale or size of a company, while its management quality indicates its soundness as a company, and technical efficiencies are a useful measure to evaluate how a company's production processes scale in terms of efficiency against other players in the industry [7,8]. Thus, we selected an average-sized, sound, technically efficient company as the representative company. The authors in [9] published a survey targeting Korean furniture companies that collated the responses of 236 companies and evaluated their management quality and technical efficiencies. They left the full results and analysis open, which assisted us in selecting a representative company for this research.

After selecting a representative company, we collected data relevant to the production of furniture, using MDF and its procurement amount, by the company through an interview with the company's CEO. All furniture manufacturers procure pieces of MDF with a specific thickness, a specific width, and a specific height (for example, *thickness* $\times$ *width* $\times$ *height* $=$ $28\,\text{mm} \times 2400\,\text{mm} \times 1200\,\text{mm}$), and produce furniture by cutting those pieces into various products sizes. This MDF cutting process generates leftovers, and reducing them saves production costs and helps protect the environment.

With the data on MDF use, we seek the minimum amounts of MDFs, required to match the 2016 furniture production levels by employing the bin packing/cutting stock problem, which is an essential step in production planning for the wood, glass, paper, and cloth industries [10,11]. This research focused on the two-dimensional bin packing/cutting stock problem, as furniture production allocates a set of rectangular items to larger rectangular MDF with standardized sizes, by minimizing leftovers. Since the 1960s, a considerable amount of literature has been published on the one-dimensional bin packing/cutting stock problem [12]. The authors in [13,14] introduced the first model for this problem, [15] extended the first mathematical model including upper bounds on the number of leftovers, and [16] suggested the L.P. model for the one-dimensional bin packing problem. The first attempt to model two-dimensional bin packing/cutting stock problems was also made by [17], who proposed a column generation approach. The authors in [18] summarized the state-of-the-art of bin packing problem until 2006, and presented a new approach to bin packing problem which was different from previous approaches. Recently, [19] suggested a simple linear model for two-dimensional problems. As the problem has been proven to be strongly *NP-hard*, many heuristic and metaheuristic approaches have also been proposed. This article works to solve the two-dimensional bin packing problem for a representative company in the Korean furniture industry by using a commercial solver, and our trial was successful because of the advances in computer technology and solver capacity.

In our paper, we compare the optimal solutions from the two-dimensional bin packing problem, with real 2016 production results from the representative company, evaluated the managerial and environmental differences in the use of MDF, apply our findings to the entire Korean furniture industry, and suggest the direction of improvement.

In the "Background and Problem Setting" section, we discuss the production process and characteristics of MDF, and consider the mathematical model of the two-dimensional bin packing/cutting stock problem used in this research. The summary of the empirical data, and the explanation of some issues in the data, are presented in the "Empirical Data" section. In the "Results and Discussion" section,

we illustrate the results of empirical analysis for the representative company, compare those results with real production results from 2016, and suggest new managerial insights. Finally, the "Conclusions" section summarizes the paper.

## 2. Background and Problem Setting

In this section, we discuss the production process and characteristics of MDF, which is the main material for the production of furniture, and consider the two-dimensional bin packing problem developed by [19], which is used to solve the optimization problem for the representative company of Korea furniture industry.

### 2.1. Medium Density Fiberboard (MDF)

As noted above, wood-based panels, used for furniture production, are classified into natural wood, plywood, PB, and MDF. Natural wood is wood itself, which is not processed at all except for washing and cutting. The use of natural wood in furniture production has decreased since 1970, because it is very expensive and is not easily processed into various shapes, although furniture made with natural wood is aesthetically pleasing and free from environmental hormones, such as formaldehyde [6]. Plywood consists of thin plates of wood glued together so that the wood grain of each layer is orthogonal to its neighbours. It solves the problem that a single slate is easy to break along the grain and is easy to shrink and expand due to moisture. Printed plywood with a pattern printed on the surface or fancy plywood with a melamine resin plate are also made and widely used in furniture, building materials, and sports fields as well as construction. Urea resin and phenol resin are used as adhesives [20]. More than 90% of plywood is used for producing dies for construction and house interiors rather than furniture, as it is relatively expensive, and not easily processed and changed.

Both PB and MDF are wood-based panels made by grinding wood, mixing the ground wood with adhesive, and hardening it with heat and pressure. While they are organic and economical because they are produced with discarded or recycled wood, and easily processed into various shapes, panels of PB and MDF are easily deformed by moisture and water. They also cause new furniture syndromes, such as atopy and respiratory disturbance because the adhesives used. The main difference between PB and MDF is that, because MDF particles are smaller and more tightly bonded than PB, it is stronger. This difference explains the fact that far more MDF is produced and used for furniture production in Korea, although they are almost homogeneous in the aspect of production cost and formaldehyde emission [6].

MDF is classified into the following four grades, based on the amount of formaldehyde emission: Super E0, E0, E1, and E2. E2 must not be used inside; E1 and E0 have an indoor use area limitation; only super E0 can be used inside freely [6]. However, because approximately 98.4% of MDF used in Korea belongs to the E2 and E1 grades [6], the amount of MDF used for furniture production should also be reduced to protect the environment and save on production costs. Table 2 summarizes the information on MDF grades.

**Table 2.** Summary of information of MDF grades.

| Grades | Average Emission ($\mu g/m^2/h$) | Ratio Used in Korea |
|:---:|:---:|:---:|
| E2 | 600 | 89.7% |
| E1 | 120 | 8.7% |
| E0 | 20 | 1.3% |
| Super E0 | 5 | 0.3% |

### 2.2. Mathematical Model of Two-Dimensional Bin Packing Problem

The two-dimensional bin packing/cutting stock problem is defined as a problem with an unlimited number of identical rectangular bins of width $W$ and height $H$, and where the objective is to allocate all

elements (items) to the minimum number of bins [11]. It can be considered as follows. We are given $n$ elements (wood boards for each type of table, desk, and bookshelf), having width $w_j$ and height $h_j$ ($j = 1, 2, \ldots, n$) and $n$ identical potential bins (MDF boards) of width $W$ and height $H$ ($k = 1, 2, \ldots, n$) though all $n$ bins may not be used to pack $n$ elements. Each bin can have multiple levels and the number of them indicates how many elements are packed into, from the bottom of the bin to the top of it. If an element is packed into the right side to an element in the same level, it does not affect total number of levels of the bin. $n$ potential levels are also assumed to be given ($i = 1, 2, \ldots, n$) though all $n$ levels may not be used to pack $n$ elements because multiple elements can be packed into a level and the number of element packed first into the leftmost-bottom of a level becomes the number of the level. If a bin has a specific number of levels, it means that no element is packed to the top of elements packed into the number of levels of the bin. We used the mathematical model developed by [19] in which elements are packed by levels [11,19]. They assume that (1) the leftmost element packed in each level is the tallest one in the level, (2) the bottom level packed in each bin is the tallest one in the bin, and (3) the elements are sorted so that $h_1 \geq h_2 \geq \ldots \geq h_n$ [19]. See [19] for the details on level packing. The packing process consists of two steps: Packing elements into levels and packing levels into bins. Each level has its own base width $w_i$ and its own height $h_i$, which are the width and the height of element packed first in the level, as the element packed first into a level is the highest of all elements packed, its height becomes the height of level $i$ and $W - w_i$ becomes the remaining width space for other elements packed into the right side of the element packed first in level $i$. Each bin also has its own base height $h_k$ which is the height of level packed first in the bin and $H - h_k$ becomes the remaining height space for other levels packed into the top of the level packed first in bin $k$. We can define element packing variable $x_{ij}$ to be 1 if element $j$ is packed into level $i$ and to be 0 otherwise for all $j$ which is greater than or equal to $i$ and level packing variable $z_{ki}$ to be 1 if level $i$ is allocated to bin $k$ and to be 0 otherwise for all $i$ which is greater than or equal to $k$. Moreover, the variable $y_i$ is defined as a variable indicating whether element $i$ is packed first into level $i$ or not ($= x_{ii}$) and variable $q_k$ is defined as variable indicating whether level $k$ is allocated into bin $k$ or not ($= z_{kk}$). As noted, when level $i$ is assigned first to the left-bottom of bin $k$, $i$ becomes equal to $k$, which means additional new bin $k$ (MDF board) is used to include levels which are greater than $k$ to the top or right of level $k$ and we have to add 1 ($z_{kk} = q_k = 1$) to total number of bins (MDF boards)

The mathematical model of [11] is as follows:

$$\underset{x_{ij}, z_{ki}, y_i, q_k}{Min} \sum_{k=1}^{N} q_k, \tag{1}$$

subject to,

$$\sum_{i=1}^{j} x_{ij} = 1 \text{ for all } j, \tag{2}$$

$$\sum_{j=i+1}^{n} w_j x_{ij} \leq (W - w_i) y_i \text{ for all } i, \tag{3}$$

$$\sum_{k=1}^{i} z_{ki} = y_i \text{ for all } i, \tag{4}$$

$$\sum_{i=k+1}^{n} h_i z_{ki} \leq (H - h_k) q_k \text{ for all } k \tag{5}$$

$$x_{ij}, z_{ki}, y_i, q_k \in [0, 1] \text{ for all } i, j, k, \tag{6}$$

Objective function (1) implies minimizing the number of bins (MDF boards) used. Constraints (2) imply that each element is packed exactly once into level $i$ ($\leq j$). Constraints (3) mean that the sum

of widths of all elements packed into a level should be less than, or equal to, the width of the level. Constraints (4) impose that each level should be allocated exactly once in a bin $k$ ($\leq i$). Constraints (5) mean that the sum of heights of all levels allocated into a bin should be less than, or equal to, the height of the bin. As noted, using the above mixed integer linear program model, we solve the two-dimensional bin packing/cutting stock problem of the representative company of Korea furniture industry.

Thus, we employed CPLEX version 12 embedded in the GAMS (General Algebraic Modelling System) to solve the model. CPLEX uses a branch and cut algorithm to solve the mixed integer programming problems, which is the most efficient algorithm [21,22], because this problem may be a very complex, mixed-integer problem.

## 3. Setting of Empirical Study

We selected a representative company in the Korean furniture industry and implemented an empirical study with its 2016 production data, as noted. The representative company was selected based on 2016 sales, Korea National Productivity Award Index (KNPAI) evaluated by Malcolm-Baldridge management quality model [23], and technical efficiencies (TE) of 236 furniture production companies in Korea, obtained from the survey of [9]. This representative company can be considered to be a benchmark for other companies.

The 2016 sales, KNPAI, and technical efficiencies from the survey of [9] can be summarized, as presented in Table 3. The KNPAI is measured on a 1000 point-full scale, and most companies are located on the scale between 0 and 500, industry leaders are located between 500 and 700, and world class companies are located over 700 [23]. Technical efficiencies are measured in three ways: Variable returns to scale (VRS), constant returns to scale (CRS), and scale efficiency (SE) as the ratio of technical efficiency based on CRS to technical efficiency based on VRS.

**Table 3.** Summary of the KNPAI and technical efficiencies.

|  | Sales (US $) | KNPAI | TE Based on VRS | TE Based on CRS | Scale Efficiency |
|---|---|---|---|---|---|
| N | 236 | 236 | 236 | 236 | 236 |
| Min | 53,619 | 316.60 | 0.3264 | 0.3172 | 0.4809 |
| Max | 604,906,989 | 634.14 | 1.0000 | 1.0000 | 1.0000 |
| Mean | 14,676,584 | 494.42 | 0.7763 | 0.7206 | 0.9316 |
| Standard Deviation | 52,969,075 | 46.84 | 0.2037 | 0.2067 | 0.1089 |

We selected a company whose sales are closest to the mean sales, it is an industry leader by KNPAI, and what is technically efficient to be the representative company. Its sales are (US) $14,712,344.88, which includes all kinds of furniture such as chairs, steel cabinets, sofas etc., as well as tables, desks, and bookshelves. Its KNPAI is 537.77, which means it is technically efficient. The company produces tables, desks, and bookshelves with 130 different size-MDF boards by cutting $t : 28\,\text{mm} \times w : 2400\,\text{mm} \times h : 1200\,\text{mm}$ raw MDF boards. Each size of MDF board is used by one or more than 1. For example, the amount of $w : 2400\,\text{mm} \times h : 1200\,\text{mm}$ MDF board used is 4, the amount of $w : 1800\,\text{mm} \times h : 800\,\text{mm}$ MDF boards used is 196, and the most amount of boards is 799 of $w : 1200\,\text{mm} \times h : 600\,\text{mm}$. In this way, a total of 5152 MDF boards are used to produce tables, desks, and bookshelves. The company bought 5103 $t : 28\,\text{mm} \times w : 2400\,\text{mm} \times h : 1200\,\text{mm}$ MDF boards at the procurement cost of $124,396.98 ($= 5,103 \times \$24.38$) in 47 order placement times and at the transportation cost of $4181.49 ($= 47 \times \$88.97$). Thus, we solved a two-dimensional bin packing problem with 5152 elements, potential levels, and potential bins by using the model of [19].

## 4. Results and Discussion

As noted in the "Backgrounds and Problem Setting" section, we solved a two-dimensional bin packing/cutting stock problem with the data of the representative company by using CPLEX 12 embedded in GAMS. The results are as follows.

1898.345 s were spent in solving this two-dimensional bin packing problem of 26,548,256 binary variables and 20,607 constraints with 32 GB ram on a 3.3 Ghz computer. For the interpretation of the optimal decision variable values, we used a bin as an example, because the entire result is too big to be shown in this paper. Let us take a look bin 3275. In the optimal solution, because $z_{3275,3275} = 1$ and $z_{3275,4832} = 1$, two levels of 3275 and level are packed into bin 3275. Two elements of 3275 and 3318 are packed into level 3275 and element 4832 is packed into level 4832 at the top of level 3275. This implies that level 3275 is allocated first into bin 3275 and level 4832 is allocated on the top of level 3275 into bin 3275. Moreover, we can say that element 3275 and element 3381 are packed into the right side of elements 3275 in level 3275 of bin 3275 and element 4832 is packed into level 4832 (the top of level 3275) in bin 3275 from $x_{3275,3275} = 1$, $x_{3275,3318} = 1$, and $x_{4832,4832} = 1$. Because $q_k = z_{kk}$, $q_{3275}$ becomes 1 while $q_{4832}$ becomes 0 and because $y_i = x_{ii}$, $y_{3275}$ and $y_{4832}$ become 1 while $y_{3318}$ becomes 0. Figure 1 illustrates this result.

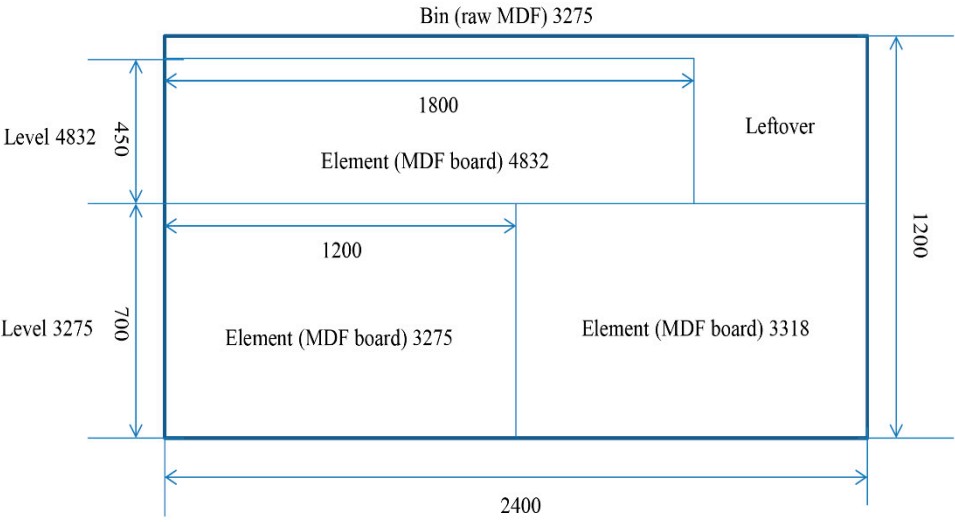

**Figure 1.** Interpretation of level packing/cutting result for bin (raw MDF) 3275.

A total of 3186 MDF out of 5152 potential boards is the minimum number of MDF boards that can produce the 5152 boards needed to produce furniture for this company. Because the total area of 3186 ($w$ : 2400 mm $\times$ $h$ : 1200 mm) MDF boards is 9175.68 m$^2$ and the total area of MDF boards for production is 5575.42 m$^2$, the amount of leftovers totals roughly 3600.26 m$^2$. Because the price and transportation of 2.88 m$^2$ is $24.38 and $0.89, the total purchase cost of MDFs is $80,500.36, and the total cost of leftovers is $31,585.94. The average emission of formaldehyde of the optimal production plan is 1,752,480 µg/m$^2$/h.

In 2016, because the company bought 5013 MDF boards and it had 460 in inventory at the end of the year, it used 4553 boards to produce 5152 products. The total area of MDF boards used was 13,112.60 m$^2$, and leftovers were 7537.18 m$^2$. The total purchase cost of MDF boards was $128,578.47 and the total cost of leftovers was $63,797.12. The average emission of formaldehyde of the present production plan is 2,504,400 µg/m$^2$/h.

Comparing the results from the optimal production plan with those from the 2016 real production plan, we found that the representative company spent more on leftovers by $32,211.17 and emitted more formaldehyde by 751,920 µg/m$^2$/h in 2016, than in the optimal production plan. For the extension of the above result to the entire industry, if it came from statistical analysis to the sample of [9], it is possible for the population of this research to follow normal distribution asymptotically and to estimate the robust industry-wide result by using it [24]. However, because it came from the representative company, it may not be reasonable to do it, although it is clear that the model has value in the results presented for the representative company, and that substantial savings are possible

industry-wide. There are limitations. The above result are the only connection in obtaining some insight to industry-wide results, even though the findings are dim, we must also assume that the results are an average outlook of the industry, as the 2016 sales of the representative company are very close to mean of sample of 236 companies. Keeping these factors in mind, if we extend the results to all of 11,572 companies a total \$372,747,709.89 will be saved resulting in an 8,701,218,240 µg/m$^2$/h reduction in formaldehyde emission. Instead, even though the result of the representative company is extended to 236 companies in the samples of [9], the production cost of \$7,610,733.95 will be saved and 177,453,120 µg/m$^2$/h of formaldehyde emission will be reduced.

In conclusion, the Korean furniture industry can reduce production cost and formaldehyde emission substantially by following the optimal production plan obtained by applying the two-dimensional bin packing/cutting stock model.

## 5. Conclusions

This research was motivated by the possibility of production cost savings and environmental protections by reducing MDF leftovers in the furniture production process in the Korean furniture industry. MDF is broadly used to produce furniture such as tables, desks, and bookshelves as it is cheaper than plywood, resistant to shocks, and easily processed. However, it has weaknesses in that it is less resistant to moisture and water, and emits formaldehyde in the air because it includes some adhesives through its production process. Thus, if furniture producers can use the minimum amount of MDFs within their production plans, they can both save their own production cost and contribute to environmental protection. An approach in this effort is to utilize a bin packing/cutting stock model. Specifically, the two-dimensional bin packing/cutting stock model is suitable for the furniture industry because the top boards of tables, desks, and shelves of bookshelves are mainly measured by their width and height.

Because the bin packing/cutting stock problem has been proven to be strongly *NP-hard*, research has focused on developing heuristic algorithms to solve the problem rather than applying the problem model to production processes of industries. However, we can now apply the bin packing/cutting stock model to industries, such as furniture, paper, glass, and clothes by using advanced computer technology.

Thus, this research tries to select a representative company in the Korean furniture industry based on sales, KNPAI, and technical efficiency, collect its production data using MDF and MDF procurement, obtain the optimal production and procurement plan by solving a two-dimensional bin packing/cutting stock problem, and compare the results with real production and procurement plans. This research suggests an opportunity to reduce the amount of MDF boards used to produce tables, desks, and bookshelves and, as a result, substantially reduce the amount of MDF leftovers. We also show the possibility of reducing formaldehyde emission considerably. The model and results from this study can be spread to other industries and uses, such as the paper, wood, glass, and steel industries.

**Funding:** This research was funded by the 2015 Incheon National University Intramural Research Fund.

**Acknowledgments:** The following are acknowledgement for willingly sharing their survey and research results to this research; S.L., Y.J., S.K., J.Y. and B.C.

**Conflicts of Interest:** The authors declare no conflict of interest.

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
