# Peer review of "Production Planning to Reduce Production Cost and Formaldehyde Emission in Furniture Production Process Using Medium-Density Fiberboard"

_processes, doi:10.3390/pr7080529_

Round 1

Reviewer 1 Report

This paper applies a two dimensional bin packing model to a problem faced in furniture production.  The author shows that this model application can substantially reduce both cost and waste in the production of furniture using medium density fibreboard, which emits formaldehyde (which is potentially hazardous).  It is a fairly straightforward application of a well-known model using a commercial solver.  However, the results have practical importance in terms of production cost, efficient usage of natural resources, and environmental impact.  What is difficult to understand is the total environmental impact.  The company produces the same amount of furniture, presumably, and the environmental impact seems to be a result of the amount of furniture produced, and not the number of boards used in production.  This should be clarified.  In addition, as I note in a comment in the attachment, the generalization of the results of the problem instance solved to proportional savings throughout the entire industry is a claim made without reasonable basis.  I think it is clear that the model has value via the results presented for the representative company, and that substantial savings are possible industry-wide.  But to assign a number to it without a deeper industry-wide analysis is not reasonable. 

Author Response

Thank you for your great comments to my manuscript. They were very helpful to improve it. In order to respond your comments I re-arranged them to #1 and # 2 and also I reflected all your suggestions in pdf file to my revised manuscript.

1. The company produces the same amount of furniture, presumably, and the environmental impact seems to be a result of the amount of furniture produced, and not the number of boards used in production. This should be clarified.

>> Yes, your comment is right. l added a paragraph like the following to introduction.

The demand amount of furniture produced using MDF cannot be controlled by furniture manufacturers. Customers or environmentalists will do that. So, manufacturers must take the demand as decided and they can contribute to reducing formaldehyde emitted into air only by reducing the amount of MDF used in their production process as well as producing the amount of furniture which customers want. This means that reduction in total amount of MDF existing in Korea (either as raw material or as furniture) results in the reduction in total amount of formaldehyde in the air of Korea.

2. In addition, as I note in a comment in the attachment, the generalization of the results of the problem instance solved to proportional savings throughout the entire industry is a claim made without reasonable basis. I think it is clear that the model has value via the results presented for the representative company, and that substantial savings are possible industry-wide. But to assign a number to it without a deeper industry-wide analysis is not reasonable.

>> Ok, good point. So I tried to delete this part at first but still because other reviewers were interested in this part I added the following paragraph in p.7 in order to justify it. I hope that you will be satisfied with it.

For the extension of the above result to whole industry, if it came from statistical analysis to the sample it is possible for the sample of this research to follow normal distribution asymptotically and to estimate the robust industry-wide result by using it (Hogg et al. ()). However, because it came from a company, it may not be reasonable to do it though it is clear that the model has value via the results presented for the representative company, and that substantial savings are possible industry-wide. In this situation, admitting that the above result is still the only way to obtain some insight to industry-wide result,  even though very dim, and assuming that it is the average result because the 2016 sales of the representative company is very close to mean of sample of 236 companies, as noted in introduction section...

Reviewer 2 Report

First of all, very interesting work.

I'd just like to comment on a few minor changes:

In section 2.2. variables of the linear model are defined. For example, zki  checks whether level i is placed in bin k (for i greater than k). Then qk is defined as zkk. My cuestion is, if i can be equal as k, I think it is important to emphasize it.

The phrase in line 239 where this text appears:

        "two levels of level 3275 and level 4832 are allocated into bin 3275"

        I understand what this text is trying to explain but I think you kind of misspelled that phrase in         particular. Perhaps it would be interesting to rewrite it.

Regards

Author Response

Thank you for your great comments to my manuscript. They were very helpful to improve it. In order to respond your comments I re-arranged them to #1 and # 2.

In section 2.2. variables of the linear model are defined. For example, zki checks whether level i is placed in bin k (for i greater than k). Then qk is defined as zkk. My question is, if i can be equal as k, I think it is important to emphasize it.

>> I added the following sentences in p.5 to respond to this comment.

As noted, when level i is assigned first to the  leftmost-bottom of bin k, i becomes equal to k, which means additional new bin k (MDF board) is used to include levels which are greater than k to the top or right of level k and we have to add 1 () to total number of bins (MDF board).

Also, I had to amend "all j which are greater than i" into "all j which are greater than or equal to i" and do the same thing to i and k in line 198 and 199.

The phrase in line 239 where this text appears: "two levels of level 3275 and level 4832 are allocated into bin 3275" I understand what this text is trying to explain but I think you kind of misspelled that phrase in particular. Perhaps it would be interesting to rewrite it.

>> I am sorry but I couldn't find any substantial misspelling in the sentence. If I can rewrite it, the following may be the best.

The two levels of 3275 and 4832 are packed into bin 3275. Two elements of 3275 and 3318 are packed into level 3275 and element 4832 is packed into level 4832 at the top of level 3275.

Reviewer 3 Report

GENERAL COMMENT

This paper focuses upon the reduction of production costs and formaldehyde emission from wood based materials in furniture production. The topic is very interesting from industrial practices point of view. The topic is of current interest and the testing reported could produce valuable outcomes, anyway the research presents the following issues:

Keywords

·  To the keywords Author should add another wood based materials e.g. PB, plywood.

 Introduction

·  There is a lack of information about factors which influence on the formaldehyde emission e.g. during MDF, PB, plywood process production (kind and amount of polycondensation resins, kind and reactivity of hardeners). Author should describe formaldehyde emission from cladding materials (e.g. veneers, melamine papers) and adhesives commonly used for veneering process. Those factors influence generally on the emission process.

· Author wrote that MDF and PB consist formaldehyde which is a toxic substance. Nowadays producers of wood based materials used resins with lower emission level fulfilling requirements super E0 class.

· Author described data about production of different wood based materials in Korea and other countries. Collecting such data in the table will be better for reading the paper.

· Information given at the end of introduction (what will be in next part of the paper) are not necessary in the scientific paper!

·  Why Author did not give the aim of the paper?

 Backgrounds and problem setting

·  Author describe mainly MDF. In the production of furniture elements are using PB, plywood and solid wood. Those materials consist some agents which influence on the formaldehyde emission. There is a lack of information about the finishing process. In industrial practice there cannot be used only wood based material without cladding/finishing materials.

· Why Author had used this mathematical model for formaldehyde emission estimation from furniture elements? In this formula other factors should be given (kind and amount of polycondensation resins, kind and reactivity of hardeners, kind of cladding materials and emission).

 Setting of Empirical Study

· There is a lack of the information about technological process of the production of furniture elements – only possibilities of using (chairs, sofas, tables, desks, etc.).

· Why Author analyzed only MDF? What about another materials PB, plywood or solid wood?

 Results and Discussion

· Analysis of obtained results was given acc. to the mathematical formula.

· The two-dimensional bin/packing/cutting stock problem cannot be solved only by cutting of bigger elements into smaller. Producers of furniture elements can introduce to the practice wood based materials with the lower density. This factor in the mathematical formula can help solve this problem.

 Conclusions

· Conclusions only concern MDF. What about another materials (PB, plywood …)? When Author have only data for MDF he should change the title of the paper!

 References

· There is a lack of papers of the European Authors. Author should add some papers.

Paper can be published after major changes and additions.

Author Response

Thank you for your great comments to my manuscript. They were very helpful to improve it. In order to respond your comments I re-arranged them from #1 to #13.

Keywords:

To the keywords Author should add another wood based materials e.g. PB, plywood.

>> I just change the title of this paper following your suggestion into "Production Planning to Reduce Production Cost and Formaldehyde emission in Furniture Production Process using Medium-density Fiberboard" like your suggestion in the comment for conclusion. Only MDF is target of this paper. The reason is described in the response to comment 7.

Introduction:

There is a lack of information about factors which influence on the formaldehyde emission e.g. during MDF, PB, plywood process production (kind and amount of polycondensation resins, kind and reactivity of hardeners). Author should describe formaldehyde emission from cladding materials (e.g. veneers, melamine papers) and adhesives commonly used for veneering process. Those factors influence generally on the emission process.

>> I just described the reasons why this paper focuses only on MDF rather than other wood-based materials in introduction (from line 41 to line 94) and also in background and problem setting section. To my knowledge, MDF emits formaldehyde mainly and mostly because of adhesives used to amalgamate ground wood because no chemical is used during grinding and hardening processes and this paper is interested in the amount of MDF used to produce furniture and formaldehyde emission from MDF itself but is not interested in other materials used in furniture production process. I think that I made this point clear sufficiently. However, I added the following sentences and changed the location of some sentences between line 64 and 77 in introduction section to reflect this comment and make the point clearer.

Both PB and MDF emit formaldehyde mainly and mostly because of adhesives used to amalgamate ground wood because chemicals are rarely used during grinding wood process and hardening process. In other words, the main source of formaldehyde emitted from PB and MDF is adhesives which are used to amalgamated ground wood to make it into the shape of board.

Author wrote that MDF and PB consist formaldehyde which is a toxic substance. Nowadays producers of wood based materials used resins with lower emission level fulfilling requirements super E0 class.

>> I don't know the source of your data but in line 156-157 and table 2 (table 1 in originally submitted manuscript) I presented a statistic that most of MDF used for furniture production in Korea is still E1 and E2.

So I added a sentence in the lines " although they encourage the use of top-level MDF or the reduction of MDF use [18] and producers of MDF are encouraged to use adhesives and resins with lower emission level of formaldehyde fulfilling the requirement of super E0 class, MDF of other classes are still used broadly [18]" .

Author described data about production of different wood based materials in Korea and other countries. Collecting such data in the table will be better for reading the paper.

>> Ok, good suggestion. So, I created table 1 to do what you requested and re-numbered the next tables.

Information given at the end of introduction (what will be in next part of the paper) are not necessary in the scientific paper!

>> Yes, this comment is controversial. Somebody agrees with you but others do not. I will talk about this with editors and modify it according to your opinion and their opinion.

Thank you.

Why Author did not give the aim of the paper?

>> Good comment but I think that it is given to line 89 to 92 which starts with "This research focuses on helping..." and then I described what will be done to achieve the aim of goal.

Backgrounds and problem setting:

Author describe mainly MDF. In the production of furniture elements are using PB, plywood and solid wood. Those materials consist some agents which influence on the formaldehyde emission. There is a lack of information about the finishing process. In industrial practice there cannot be used only wood based material without cladding/finishing materials.

>> Because the target wood based material of this paper is MDF, I just described MDF mainly. However, I also described plywood, solid wood and PB presenting why I selected MDF as target material of this research. Plywood and solid wood are not included in the analysis of this paper for the reasons presented from line 41 to 45 and from line 154 to 164. The reason why MDF is selected over PB was also described from line 98 to line 101. So, I did not need to describe the production process of plywood, solid wood, and PB. Moreover, as noted, this paper is interested only in the amount of MDF itself and emission from adhesives in MDF though surely there are some additional emission factors in the production process. Hopefully, future research may include them. Thank you again.

Why Author had used this mathematical model for formaldehyde emission estimation from furniture elements? In this formula other factors should be given (kind and amount of polycondensation resins, kind and reactivity of hardeners, kind of cladding materials and emission).

>> Frankly speaking, the mathematical formula is based on the classical one to calculate the minimum amount of material (here, MDF) to produce the pre-determined amount of products (here, furniture elements). After that, emission is estimated by using emission rate of each level of MDF in table 2, average usage of each level of MDF in Korea, and result from the mathematical formula. This process was described through paper.

Setting of Empirical Study:

There is a lack of the information about technological process of the production of furniture elements – only possibilities of using (chairs, sofas, tables, desks, etc.).

>> The reason why the production process for each furniture elements is not described is that this paper focuses on saving production cost and formaldehyde emission through reducing the amount of MDFs used to produce furniture elements. Thus, the amount of MDF required for the production of each furniture element is important but production process of furniture elements is not important.

Why Author analyzed only MDF? What about another materials PB, plywood or solid wood?

>> The target material of this paper is MDF because of the reasons listed in the responses to comment 7. As noted, I modified title for the convenient of understanding.

Results and Discussion:

Analysis of obtained results was given acc. to the mathematical formula. The two-dimensional bin/packing/cutting stock problem cannot be solved only by cutting of bigger elements into smaller. Producers of furniture elements can introduce to the practice wood based materials with the lower density. This factor in the mathematical formula can help solve this problem.

>> I got your point but because the mathematical formula is based on the classical model to solve bin packing problem as noted in the response to comment 8 and this research excludes the other wood based materials focusing on only MDF for the reasons discussed in comments 7 and 10, It does not look problematic to solve this problem and also looks unnecessary for my math formula to be modified.

Conclusions:

Conclusions only concern MDF. What about another materials (PB, plywood …)? When Author have only data for MDF he should change the title of the paper!

>> Yes, I just focused on MDF because of the reasons written in the responses to your above comments (#7, #10). I changed title as noted in the response to comment 1.

References:

There is a lack of papers of the European Authors. Author should add some papers.

>> At my glance, 5 papers ([2], [4], [13], [14], and [19]) might be written by European authors. They take more than 20% of all references. Though I think that they are enough, I added the following couple papers written by European authors and sentences to introduction section:

[24] suggested L.P. model for one-dimensional bin packing problem.

[25] summarized the state-of-the-art of bin packing problem until 2006 and presented a new approach to bin packing problem which was different from previous approaches

Valério de Carvalho, J. LP models for bin packing and cutting stock problems. Euro. J. of Oper. Res. 2002, 141(2), 253–273. Wäscher, G.; Haußner, H.; Schumann, H. An improved typology of cutting and packing problems. Euro. J. of Oper. Res. 2007, 183(3),1109–1130.

Round 2

Reviewer 3 Report

The reviewer answer and comments.

Keywords

Author changed the title. In this case there is no need to add another keyword.

Introduction

Author wrote that MDF consist formaldehyde which is a toxic substance. Nowadays producers of wood based materials used resins with lower emission level fulfilling requirements super E0 class. This information is acc. to the IKEA requirements (mainly in Europe). Author created the table 1 – good answer for the reviewer proposal. Author answered for all questions.

Backgrounds and problem setting

Author changed the title. In this case everything is ok. Author had used only basic mathematical model for formaldehyde emission estimation from furniture elements. In the future (in next papers) this formula should consists other factors (kind and amount of polycondensation resins, kind and reactivity of hardeners, kind of cladding materials and emission). Advice for the future!

Setting of Empirical Study

Author modified the title for the convenient of understanding.

Results and Discussion

Analysis of obtained results was given acc. to the mathematical formula. The two-dimensional bin/packing/cutting stock problem cannot be solved only by cutting of bigger elements into smaller. Producers of furniture elements can introduce to the practice wood based materials with the lower density. This factor in the mathematical formula can help solve this problem.

 Conclusions

Author changed the title. In this case conclusions concerning MDF are correct. I know that mathematical formula is based on the classical model. The information about properties of MDF (e.g. density) are better for deeper analysis. There is a reviewer advice for the next papers!

References

Author added a few papers of the European Authors.

Paper can be published in present form.